# Ubiquitination and degradation of NF90 by Tim-3 inhibits antiviral innate immunity

**Shuaijie Dou[1,2†], Guoxian Li[1,3†], Ge Li[1†], Chunmei Hou[1], Yang Zheng[4], Lili Tang[1], Yang Gao[1], Rongliang Mo[1], Yuxiang Li[1], Renxi Wang[1,5]\*, Beifen Shen[1], Jun Zhang[3]\*, Gencheng Han[1]\***

[1]Beijing Institute of Basic Medical Sciences, Beijing, China; [2]Anhui Medical University, Hefei, China; [3]Institute of Immunology, Medical School of Henan University, Kaifeng, China; [4]Department of Oncology, First Hospital of Jilin University, Changchun, China; [5]Beijing Institute of Brain Disorders, Laboratory of Brain Disorders, Ministry of Science and Technology, Collaborative Innovation Center for Brain Disorders, Capital Medical University, Beijing, China

**Abstract** Nuclear factor 90 (NF90) is a novel virus sensor that serves to initiate antiviral innate immunity by triggering stress granule (SG) formation. However, the regulation of the NF90-SG pathway remains largely unclear. We found that Tim-3, an immune checkpoint inhibitor, promotes the ubiquitination and degradation of NF90 and inhibits NF90-SG-mediated antiviral immunity. Vesicular stomatitis virus (VSV) infection induces the up-regulation and activation of Tim-3 in macrophages, which in turn recruit the E3 ubiquitin ligase TRIM47 to the zinc finger domain of NF90 and initiate a proteasome-dependent degradation via K48-linked ubiquitination at Lys297. Targeted inactivation of Tim-3 enhances the NF90 downstream SG formation by selectively increasing the phosphorylation of protein kinase R and eukaryotic translation initiation factor 2α, the expression of SG markers G3BP1 and TIA-1, and protecting mice from VSV challenge. These findings provide insights into the crosstalk between Tim-3 and other receptors in antiviral innate immunity and its related clinical significance.

**\*For correspondence:**
wang_renxi@hotmail.com (RW);
zhangjun@henu.edu.cn (JZ);
genchenghan@163.com (GH)

[†]These authors contributed equally to this work

**Competing interests:** The authors declare that no competing interests exist.

## Introduction

Innate immunity is the first line of host defense against viral infection. Pattern recognition receptors (PRRs), including Toll-like receptors (TLRs) and RIG-I-like receptors (RLRs), are main sensors in defending virus infection (*Chen et al., 2013*). PRR-mediated downstream signaling pathways initiating an anti-viral innate immune response is the classic anti-virus infection model (*Barbalat et al., 2011*; *McFadden et al., 2017*). Recently, studies have found that nuclear factor 90 (NF90), which is encoded by interleukin enhancer-binding factor-3, is a critical sensor for invading viruses (*Li et al., 2016*; *Shabman et al., 2011*; *Wang et al., 2009*; *Wen et al., 2014*; *Pfeifer et al., 2008*). NF90 is an evolutionarily conserved member of the double-stranded RNA (dsRNA)-binding protein family and is abundantly expressed in various mammalian cells (*Patiño et al., 2015*; *Masuda et al., 2013*). As an important antiviral pathway, NF90 recognizes virus dsRNA and triggers the formation of stress granules (SGs), which are composed of cytoplasmic particles including ribonucleoproteins, RNA-binding proteins, and translation initiation factors (*Shi et al., 2007*). Unfortunately, a growing number of virus families modulate SG formation and function to maximize replication efficiency (*White and Lloyd, 2012*). Therefore, an understanding of the precise regulation mechanisms of NF90-SG signaling for efficient viral clearance without harmful immunopathology is needed.

Upon sensing virus, NF90 induces the phosphorylation of dsRNA-activated protein kinase R (PKR) (*Li et al., 2016*; *Wen et al., 2014*). Then SGs form following PKR-mediated phosphorylation and activation of the eukaryotic translation initiation factor 2α (eIF2α), which induces the expression

of G3BP1 (*Tsai et al., 2017*) (Ras-GAP SH3-binding protein-1) and TIA-1 (T-cell intracellular antigen-1), two key markers in SGs. The activated eIF2α co-operates with other components of SGs to block virus mRNA translation. Despite acting as an important virus sensor and trigger of SG formation, how NF90 is regulated remains largely unknown.

Tim-3, encoded by the hepatitis A virus cellular receptor two gene (*Havcr2*), is an immune checkpoint inhibitor that was first identified in activated T cells. Later, Tim-3 was also found to be expressed in innate immune cells, such as dendritic cells and macrophages (*Han et al., 2013*). Establishment of Tim-3 as an exhaustion marker in immune cells of both tumors and infectious diseases makes Tim-3 an attractive target for immunotherapy similar to PD-1 (*Huang et al., 2015*), CTLA-4 (*Stamper et al., 2001*), and Siglec-G (*Rangachari et al., 2012*). Recently, a report showed that increased Tim-3 expression on immune cells in patients with coronavirus disease (COVID-19) is associated with an exhaustion phenotype (*Diao et al., 2020*). However, Tim-3 does not have an inhibitory motif within its tail (*van de Weyer et al., 2006*), and the mechanism by which Tim-3 mediates inhibitory signaling remains largely unclear. *Huang et al., 2015* showed that CEACAM1 is a heterophilic ligand of Tim-3 and is required for Tim-3 to mediate T cell inhibition and that Bat-3 acts as a safety catch, which blocks Tim-3-mediated inhibitory signals in T cells (*Rangachari et al., 2012*). Potentiating anti-infection immunity by inducing innate immune responses is a promising area of infection therapy. However, little is known about Tim-3 signaling in innate immune cells.

Ubiquitination is one of the most versatile posttranslational modifications and is indispensable for antiviral infection (*Heaton et al., 2016*). Increasing evidences suggest that ubiquitination plays important roles in various cellular processes, including cell proliferation and antiviral innate signaling. Posttranslational modification of many signaling molecules, including TRAF3/6 (*Chen et al., 2015*), RIG-I (*Chen et al., 2013*), MAVS (*Liu et al., 2017*), TBK1 (*Song et al., 2016a*), IRF3/7 (*Yu and Hayward, 2010*; *Wang et al., 2015*), and NLRP3 (*Song et al., 2016b*), involved in TLR, RLR, and NLR pathways by different types of ubiquitination plays key roles in the regulation of antiviral innate immunity. However, whether NF90, a molecule containing a ubiquitin-binding domain (domain associated with zinc fingers, DZF), undergoes ubiquitination remains unclear.

Here we found that Tim-3 was involved in innate immunity against vesicular stomatitis virus (VSV) by promoting the proteasomal degradation of NF90 via the tripartite motif-containing protein 47 (TRIM47)-mediated conjugation of K48-linked ubiquitin. To the best of our knowledge, it is the first time demonstrating ubiquitination as a new posttranslational modification mechanism of NF90. Our findings shed a new light on the Tim-3-mediated immune tolerance during infection.

## Results

### Tim-3 interacts with and inhibits NF90

To test whether Tim-3 is involved in the innate immunity against viruses, we challenged macrophages with VSV, an RNA virus widely used for investigating anti-viral immunity in both mouse and human models (*Chen et al., 2013*). Shortly after VSV challenge, the expression of Tim-3 was upregulated in macrophages (*Figure 1A and B*). The phosphorylation of Tim-3, which accounts for the Tim-3 signaling (*van de Weyer et al., 2006*), was also tested. There was a time-dependent enhancement of Tim-3 phosphorylation in HEK293T cells transfected with Tim-3 and challenged with VSV (*Figure 1C*). Tim-3 was encoded by *Havcr2*, which was inactivated to generate Tim-3 knockout (Tim-3KO) mice (*Figure 1—figure supplement 1*). To evaluate the role of VSV-activated Tim-3 in host antiviral innate immune response, macrophages from Tim-3KO mice and from wild-type (WT) mice and the macrophage cell line RAW264.7 with a knockdown of Tim-3 (si-Tim-3) were challenged with VSV. Both knockout and knockdown of Tim-3 in macrophages led to a decreased VSV load (*Figure 1D and E*). These findings suggest a negative regulatory role of Tim-3 in anti-viral innate immunity.

To find the possible mechanisms of Tim-3-mediated anti-viral immunity, we used Tim-3 pulldown and mass spectrometry to identify the proteins interacting with Tim-3. Among the candidates, NF90, an RNA-binding protein involved in anti-infection and anti-tumor immunity, received the highest score and the highest number of matched peptides (*Figure 2—figure supplement 1*). To confirm the interaction between Tim-3 and NF90, HEK293T cells were co-transfected with NF90 and Tim-3 and immunoprecipitation was conducted targeting either Tim-3 (*Figure 2A*) or NF90 (*Figure 2B*); all

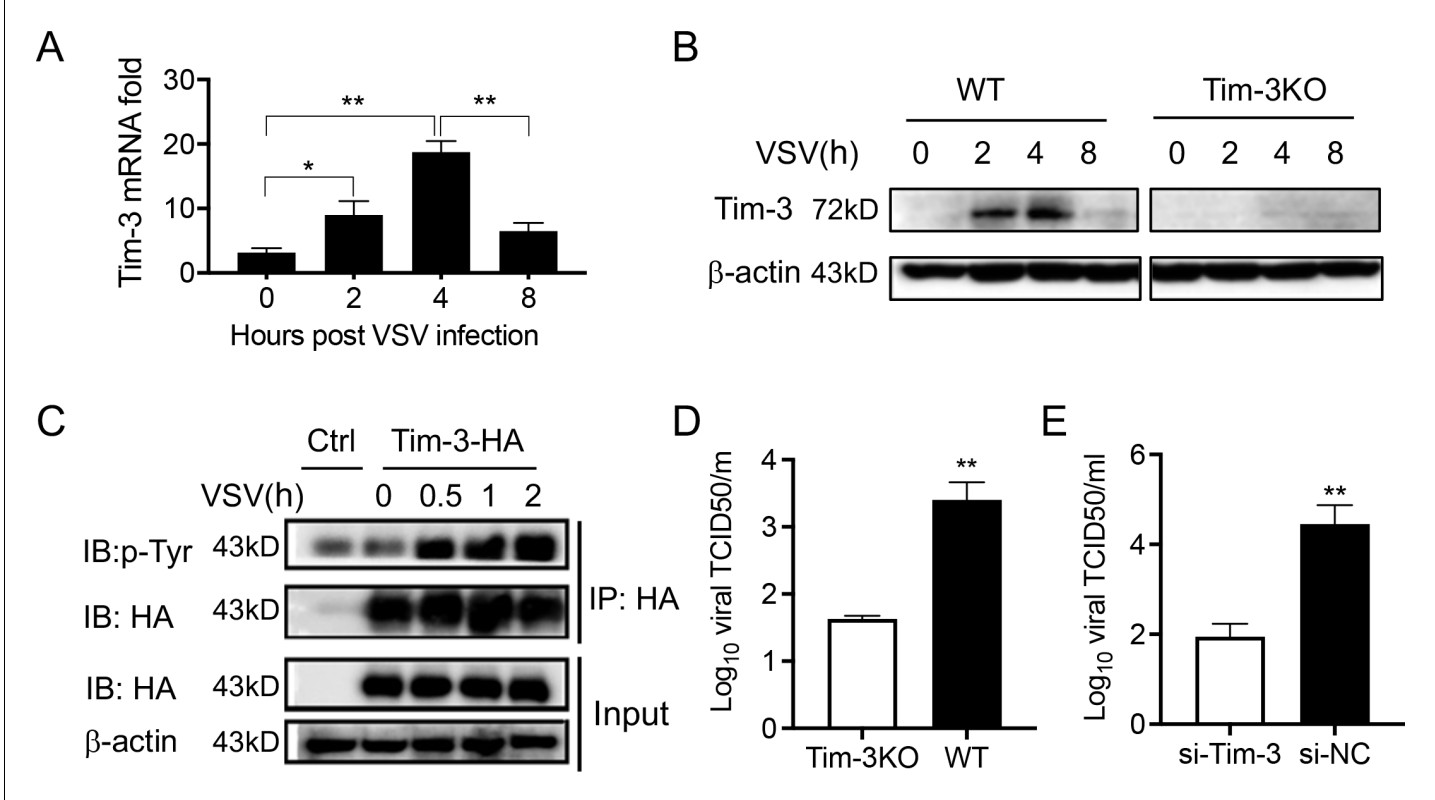

**Figure 1.** Tim-3 inhibits VSV replication in macrophages. (**A**) Peritoneal macrophages were isolated from wild-type (WT) mice and infected with vesicular stomatitis virus (VSV) for the indicated hours. Then the Tim-3 mRNA expression was examined by quantitative reverse transcription PCR (qPCR). (**B**) Peritoneal macrophages were isolated from WT and Tim-3 gene (*Havcr2*) knockout (Tim-3KO) mice and were infected with VSV for the indicated times. Then the expression of Tim-3 was analyzed by western blot analysis. (**C**) VSV infection induces tyrosine phosphorylation of Tim-3. HA-Tim-3 plasmid was transfected into HEK293T cells for 24 hr, and the cells were infected with VSV for the indicated hours. Cell lysates were immunoprecipitated with Human influenza hemagglutinin(HA) antibody and analyzed by immunoblotting for the indicated proteins. (**D and E**) Peritoneal macrophages obtained from WT and Tim-3KO mice and RAW264.74 macrophages silenced of Tim-3 (si-Tim-3) and RAW264.7 macrophages (si-NC) were challenged by VSV for 6 hr, and then the cells were harvested for VSV load analysis by 50% tissue cell infectious dose (TCID50) assay. The results shown in all panels were performed three times. *p<0.05, **p<0.01.

The online version of this article includes the following source data, source code and figure supplement(s) for figure 1:

**Source code 1.** *Flox* gene amplication.
**Source code 2.** *Cre* gene amplication.
**Source code 3.** Genotype identification.
**Source code 4.** Phenotype identification.
**Source data 1.** Tim-3 mRNA expression in macrophages in response to VSV infection.
**Source data 2.** Tim-3 protein expression in macrophage in response to VSV infection.
**Source data 3.** VSV induces Tim-3 phosphorylation .
**Source data 4.** Tim-3 signaling promotes VSV replication in macrophage .
**Figure supplement 1.** Development and identification of Tim-3 knockout mice.

confirmed the interaction between Tim-3 and NF90. The interaction sites of Tim-3 interacting with NF90 were subsequently examined. The results showed that 4Y/F mutants of Tim-3 (Y265F, Y272F, Y280F, Y281F) weakened the binding of Tim-3 with NF90, while deletion of the intracellular domain of Tim-3 (△IC) resulted in the loss of binding activity of Tim-3 with NF90 (*Figure 2C and D*). These data showed that Tim-3 interacts with NF90 through its intracellular domain, in which Y265, Y272, Y280, and Y281 play an important role. We finally evaluated the effects of Tim-3 on NF90 expression. In HEK293T cells transfected with Tim-3 or in macrophages from Tim-3 transgenic (Tim-3-TG) mice, the overexpression of Tim-3 decreased the expression of NF90 (*Figure 2E*). These results show that Tim-3 interacts with and inhibits NF90.

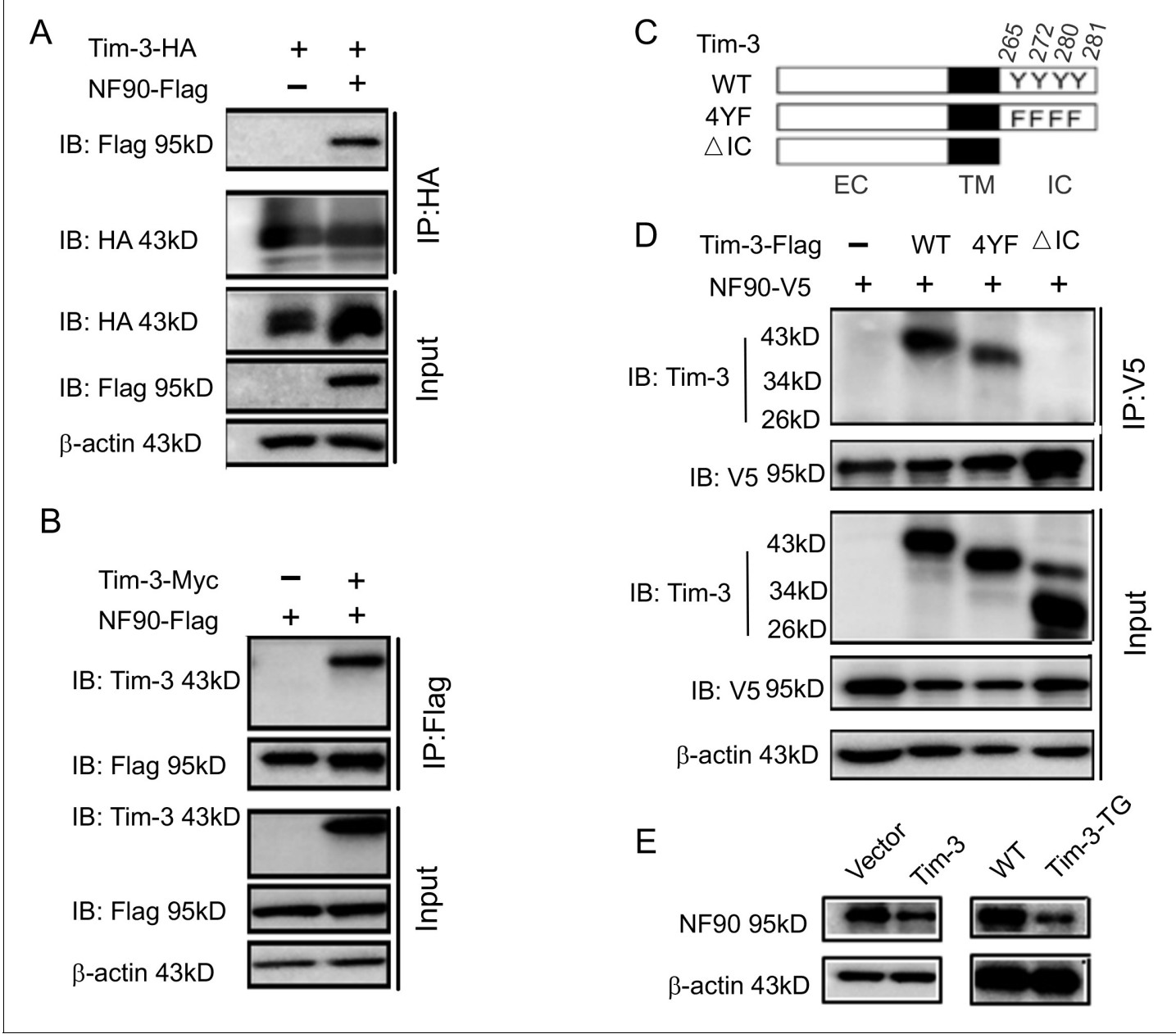

**Figure 2.** Tim-3 interacts with and inhibits NF90. (**A and B**) Protein complex of Tim-3 and NF90 overexpressed in cells. HEK293T cells were transfected with plasmids encoding HA-Tim-3, Flag-NF90, and Myc-Tim-3 for 24 hr, immunoprecipitated with HA or Flag antibody, respectively, and detected by western blot for the indicated antibodies. (**C and D**) Interaction of Tim-3 intracellular domain with NF90. Schematic structure of Tim-3 and the derivatives used are shown (**C**). Whole-cell lysis of HEK293T cells transfected with Flag-Tim-3 (WT), Flag-Tim-3 (Del), Flag-Tim-3 (4YF), and V5-NF90 was used for immunoprecipitation and immunoblotting, as indicated (**D**). (**E**) Immunoblot analysis of NF90. HEK293T cells were transfected with Tim-3 plasmid for 24 hr, and lysates were detected for NF90 expression by western blot (left). Peritoneal macrophages from wild-type (WT) and Tim-3-TG mice were lysed and NF90 protein was detected by western blot (right). The results shown in panels (**A, B, D, and E**) were performed three times. The online version of this article includes the following source data, source code and figure supplement(s) for figure 2:

**Source code 1.** Tim-3-WT were transfected into HEK293T cells.
**Source code 2.** *Candidates interacts with Tim-3.*
**Source data 1.** Tim-3 interacts with NF90.
**Source data 2.** *NF90 interacts with Tim-3.*
**Source data 3.** Tim-3 interacts with NF90 via intracellular tail.
**Source data 4.** Tim-3 inhibits NF90.
**Figure supplement 1.** MS data showing Tim-3-interacting proteins in macrophages.

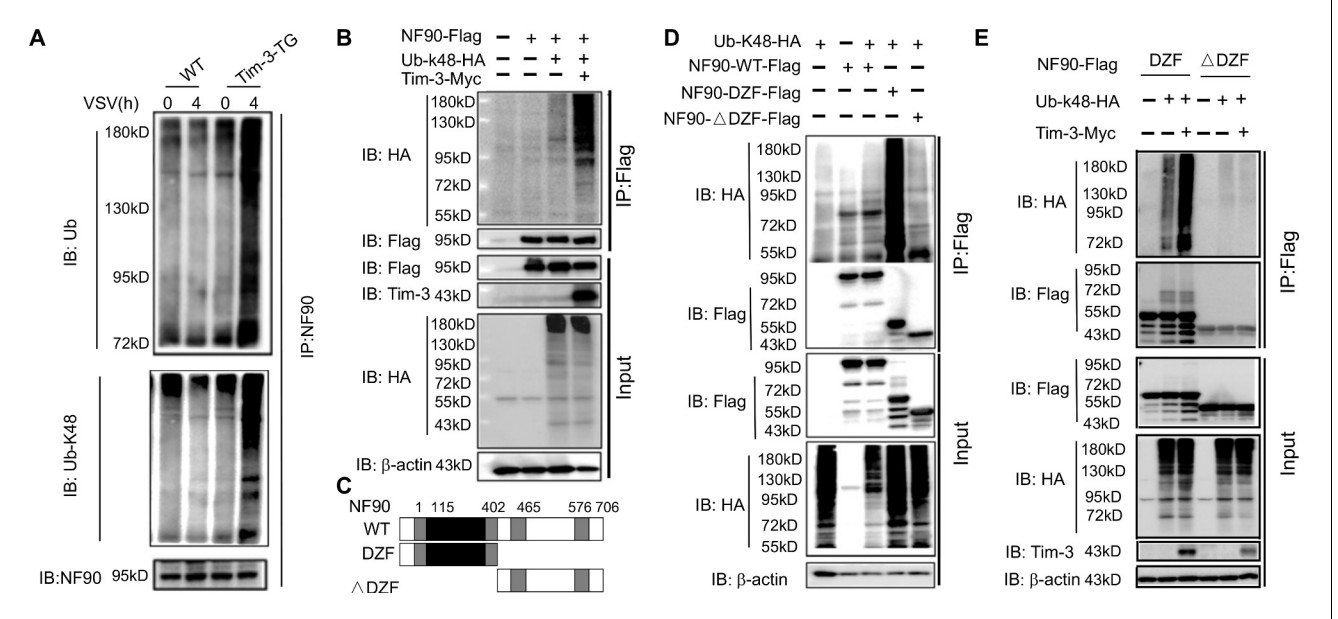

**Figure 3.** Tim-3 promotes the ubiquitination of NF90 at the DZF domain. (**A**) Tim-3 enhances the ubiquitination of NF90 in macrophages in response to vesicular stomatitis virus (VSV) challenges. Peritoneal macrophages in wild-type (WT) and Tim-3-TG mice were infected with VSV for 4 hr and cells were treated with MG132 (20 ug/ml) for 6 hr before harvesting protein lysates, followed by western blot analysis of the total Ub and K48-linked Ub of NF90 immunoprecipitated with antibody to ILF3 (NF90). (**B**) Tim-3 promotes the ubiquitination of NF90. HEK293T cells were transfected with plasmids encoding Flag-NF90 or HA-Ub-K48, Tim-3-Myc for 24 hr, treated with MG132 (20 ug/ml) for 6 hr, immunoprecipitated with Flag antibody, and then detected by western blot for the indicated antibodies. (**C**) Schematic structure of NF90 and the derivatives used are shown. (**D–E**) Tim-3 promotes the K48-Ub modification of NF90 at the DZF domain. HEK293T cells were transfected with the indicated plasmids for 24 hr and treated with MG132 (20 ug/ml) for 6 hr. The cells were then lysed, immunoprecipitated with Flag antibody, and detected by western blot using the indicated antibodies. Three independent experiments were conducted for all panels.

The online version of this article includes the following source data, source code and figure supplement(s) for figure 3:

**Source code 1.** Tim-3 mediates NF90 ubiquitination.

**Source data 1.** Tim-3 promotes ubiquitination of NF90.

**Source data 2.** Tim-3 promotes k48-linked ubiquitination of NF90.

**Source data 3.** NF90 canbe ubiquitinated at the DZF domain.

**Source data 4.** Tim-3 promotes NF90 ubiquitination at the DZF domain.

**Figure supplement 1.** Peritoneal macrophages from WT and Tim-3-TG mice were infected with VSV for 4 hr and treated with MG132 (20 ug/ml) for 6 hr.

## Tim-3 promotes the ubiquitination of NF90 at the DZF domain

Ubiquitination is one of the most versatile posttranslational modifications and is indispensable for antiviral immunity. However, whether NF90 undergoes proteasomal degradation is totally unknown. To find the mechanisms by which Tim-3 inhibits NF90 expression, we tested whether NF90 undergoes ubiquitination and if so, whether Tim-3 is involved. Macrophages isolated from WT and Tim-3-TG mice were challenged by VSV for 4 hr and the ubiquitination of NF90 was examined. Interestingly, the total ubiquitination and the k48-linked ubiquitination of NF90 were significantly increased in macrophages from Tim-3-TG mice compared to those from WT mice (*Figure 3A* and *Figure 3—figure supplement 1*). When NF90, K48-Ub, or Tim-3 was transfected into HEK293T cells, NF90 underwent K48-linked ubiquitination, which can be enhanced by co-transfected Tim-3 (*Figure 3B*). The results suggested that Tim-3 may inhibit NF90 by enhancing the K48-linked ubiquitination and degradation of NF90. We then explored the domain of NF90 for ubiquitination using constructions encoding the DZF domain, full-length NF90 or NF90 lacking the DZF domain (△DZF) (*Figure 3C*) and found that the DZF domain of NF90 was dominantly ubiquitinated (*Figure 3D*). Finally, when the plasmid of Tim-3 was co-transfected with different NF90 constructs, the data revealed that Tim-3 specially enhanced the ubiquitination of DZF (*Figure 3E*). The reason why the DZF-only construct is strongly ubiquitinated compared to the full-length construct of NF90 may lie in the conformational

hindrance in NF90 at a steady state, which needs to be demonstrated in the future. These results demonstrated that NF90 can be ubiquitinated at the DZF domain, and the process can be enhanced by Tim-3, suggesting that Tim-3 may suppress NF90 by promoting its ubiquitination and degradation.

## Involvement of TRIM47 in Tim-3-mediated degradation of NF90

To find the possible E3 ligase accounting for Tim-3-mediated NF90 ubiquitination, NF90-interacting proteins were investigated by immunoprecipitating NF90 and then performing mass spectrometry. Among the NF90-interacting protein candidates, we identified three proteins with potential E3 ligase activities. TRIM47 had the highest Mascot scores and the highest number of matched peptides (*Figure 4A*). Knockdown of TRIM47 with specific small interfering RNA(siRNA) (si-TRIM47) in macrophages increased the half-life of endogenous NF90 protein during VSV infection (*Figure 4—figure supplement 1*). To confirm whether TRIM47 promotes the proteasomal degradation of NF90, we transfected HEK293T cells with ubiquitin, NF90, and an increasing dose of TRIM47. TRIM47 induced degradation of NF90 in a dose-dependent manner, which can be blockaded in the presence of MG132, indicating a proteasome-dependent degradation (*Figure 4B*). In addition, when Tim-3 was co-transfected, it dose-dependently enhanced TRIM47-mediated degradation of NF90 (*Figure 4C*).

We then explored the possible interaction between Tim-3 and TRIM47. VSV challenge led to decreased TRIM47 expression in Tim-3KO cells compared with that in WT cells (*Figure 4D*) and increased TRIM47 expression in Tim-3 transgenic mice-derived macrophages compared with that in control cells (*Figure 4E*). These results showed that Tim-3 promotes the expression of TRIM47 in the presence of virus. The possible mechanism by which Tim-3 enhances TRIM47 expression was primarily investigated. We examined whether TRIM47 undergoes ubiquitination when Tim-3 is overexpressed, as TRIM25, an E3 ligase with a structure similar to TRIM47, undergoes ubiquitination during viral infection (*Nisole et al., 2005*; *Pauli et al., 2014*). Interestingly, when the genes encoding TRIM47, Tim-3, and Ub-K63 were co-transfected into HEK293T cells, TRIM47 underwent ubiquitination, and Tim-3 enhanced this progress (*Figure 4F*). However, the relationship between Tim-3-enhanced TRIM47 expression and Tim-3-enhanced TRIM47 ubiquitination remains to be determined. These results showed the involvement of TRIM47 in Tim-3-mediated degradation of NF90.

## Tim-3 recruits TRIM47 to the DZF domain of NF90 and Lys297 within DZF is a critical site for TRIM47-mediated K48-linked ubiquitination of NF90

To find whether Tim-3 and TRIM47 interact with each other to act on NF90, we first examined the interactions among Tim-3, TRIM47, and NF90. Different Tim-3 constructs (*Figure 5A*) or different NF90 constructs (*Figure 5B*) were co-transfected with TRIM47 into HEK293T cells. The immunoprecipitation assay showed that TRIM47 interacts with the intracellular domain of Tim-3 and with the DZF domain of NF90. These data suggest that Tim-3 recruits TRIM47 to the DZF domain via its intracellular domain while forming a complex of TRIM47 and NF90.

To confirm that Tim-3 co-operates with TRIM47 to enhance the ubiquitination and degradation of NF90, we co-transfected Tim-3, TRIM47, and ubiquitin into HEK293T cells and examined the effects of TRIM47 and Tim-3 on the ubiquitination of NF90. Overexpression of TRIM47 promotes the ubiquitination of NF90, and this process was enhanced when Tim-3 was co-transfected (*Figure 5C*). Next, we examined the ubiquitin modification site within the DZF domain of NF90 by site mutation. NF90 contains 18 lysine residues in its DZF domain. Immunoprecipitation analysis revealed that TRIM47 enhanced the ubiquitination of wild-type DZF and DZF with the K100R, K117/119R, K127R, K143R, K158R, and K224R, but not K297R, mutants in HEK293T cells (*Figure 5—figure supplement 1*). We further demonstrated that only mutations of arginine at K297R completely blocked the TRIM47-mediated ubiquitination and degradation of NF90 via a K48-mediated linkage (*Figure 5D*). In addition, when co-transfected K48-Ub, Tim-3, wild-type NF90, DZF domain, or DZF K297R with increased doses of TRIM47, we found that TRIM47 dose-dependently induced the degradation of NF90 DZF domain but not of DZF K297R (*Figure 5E*). Taken together, these data suggest that K297 is the critical residue for the ubiquitination of NF90-DZF targeted by TRIM47.

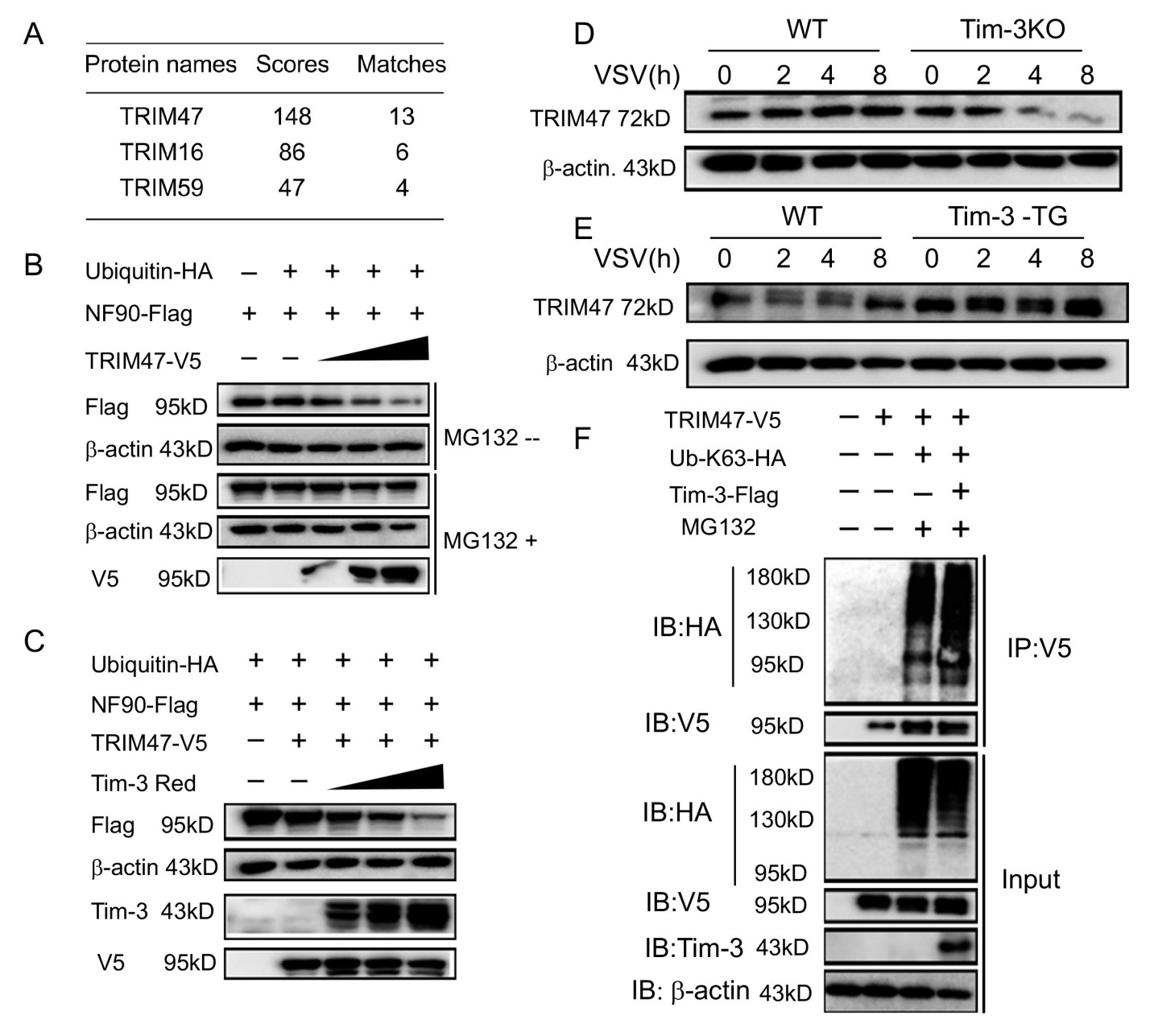

**Figure 4.** Involvement of TRIM47 in Tim-3-mediated NF90 degradation. (**A**) E3 ligases identified by mass spectrometry for top peptide hits (defined by Mascot score) associated with NF90 ubiquitination. (**B**) TRIM47 promotes NF90 degradation in a proteasome-dependent manner. Plasmids encoding Flag-NF90, HA-ubiquitin, along with increasing amounts of V5-TRIM47 (0.5, 1.0, and 2.0 ug), were transfected into HEK293T cells for 24 hr, cells were treated with and without MG132 (20 ug /ml), respectively, followed by western blot to examine the NF90 protein level. (**C**) Tim-3 accelerates TRIM47-mediated NF90 degradation in a dose-dependent manner. HEK293T cells were transfected with plasmids encoding Flag-NF90, HA-ubiquitin, and V5-TRIM47 and an increasing dose of plasmid encoding Red-Tim-3 (0.5, 1.0, and 2 ug) for 24 hr. The protein level of NF90 was examined in cells. (**D and E**) Tim-3 upregulates TRIM47 in protein levels. TRIM47 protein levels were analyzed by immunoblotting in lysates from wild-type (WT) or Tim-3KO and WT or Tim-3-TG macrophages infected with vesicular stomatitis virus (VSV) for the indicated time. (**F**) Tim-3 facilitates K63-linked ubiquitination mediated by TRIM47. Plasmids encoding HA-Ub-K63, V5-TRIM47, and Flag-Tim-3 were transfected into HEK293T cells. Cells were treated with MG132 (20 ug/ml) for 6 hr, and cell lysates were immunoprecipitated with Flag antibody and detected by western blot for K48-Ub levels. Three independent experiments were conducted for all panels.

The online version of this article includes the following source data, source code and figure supplement(s) for figure 4:

**Source code 1.** Silence of TRIM47 in macrophages.
**Source code 2.** TRIM47 promotes NF90 degradation.
*Figure 4 continued on next page*

*Figure 4 continued*

**Source data 1.** TRIM47 promotes NF90 degradation.
**Source data 2.** Tim-3 synthesizes with TRIM47 to promote NF90 degradation.
**Source data 3.** Tim-3 promotes TRIM47 expression.
**Source data 4.** Tim-3 promotes TRIM47 expression.
**Source data 5.** Tim-3promotes k63-linked ubiquitination of TRIM47.
**Figure supplement 1.** TRIM47 knockdown increases expression of NF90.

## Tim-3 deficiency enhances the formation of SGs and protects mice from VSV infection

Finally, the significance of Tim-3 in the inhibition of NF90 was investigated. As NF90 triggers the formation of SGs, we first examined whether Tim-3 regulates the downstream of NF90-SG pathway. Peritoneal macrophages were isolated from WT and Tim-3KO mice, and following VSV challenge for 2–8 hr, the expression of NF90 and the phosphorylation of PKR, eIF2α, as well as the phosphorylation of other signaling cascades including ERK and P38, were examined. The data in *Figure 6A* show that the expression of NF90 and the phosphorylation of eIF2α and PKR were dramatically increased in macrophages from Tim-3KO mice compared with those in cells from WT mice. There was no difference in p38 and ERK phosphorylation between Tim-3KO and WT cells. Meanwhile, the expression of G3BP1 and TIA-1, two markers of SGs, was also examined in above macrophages using immunofluorescence. The results in *Figure 6B and C* show that Tim-3 knockout significantly increased the expression of G3BP1 and TIA-1 at the protein level in macrophages following VSV challenges in vitro, demonstrating that Tim-3 inhibits the formation of SGs. To test the effects of Tim-3 inhibition on NF90-SG pathway in vivo, a VSV infection model was established in mice. We found that the expression of SG markers, G3BP1 and TIA-1, and VSV replication were significantly changed in spleen, lung, and peritoneal macrophages from Tim-3KO mice than those in WT mice (*Figure 7A–I*). Also, lethal VSV infections led to an increased survival rate in Tim-3KO mice compared to WT mice (*Figure 7J*) and a less severe tissue inflammation (*Figure 7K*). These data showed that Tim-3 deficiency enhances the formation of SGs in vivo and protects mice from VSV. Finally, we also examined whether the silencing of TRIM47 affects the assembly of SGs and the anti-viral immunity of macrophages. The data showed that silencing of TRIM47 with specific siRNA in RAW264.7 led to increased G3BP1 and TIA-1 expression and a decreased viral load following VSV infection (*Figure 6—figure supplement 1*), further confirming that TRIM47 acts as an up-stream regulator of the NF90-SG antiviral pathway.

The mechanisms by which Tim-3 promotes the TRIM47-mediated ubiquitination and proteasomal degradation of NF90 in viral immune evasion are summarized in *Figure 8*. Upon VSV infection, Tim-3 is activated and upregulated. The activation of Tim-3 in turn recruited the E3 ubiquitin ligase TRIM47 to the zinc finger domain of NF90 and initiated the proteasome-dependent degradation of NF90 via K48-linked ubiquitination at Lys297. The negative regulation of NF90 by Tim-3 blocked the virus triggered and NF90-SG-mediated antiviral immunity and finally led to virus immune evasion.

## Discussion

Viruses have developed many different strategies of immune evasion, for example, by downregulating or degrading virus sensors. The mechanisms by which these receptors are regulated are widely investigated in hopes of developing effective treatments. NF90 was found to play an important role in host innate immunity against various virus infections. However, the regulation of NF90 under physiopathological conditions remains largely unclear. Here we have identified a novel negative regulation mechanism of NF90, which could be employed by VSV to evade the immune attack. The VSV-activated Tim-3 in turn promotes the ubiquitination and degradation of NF90 and subsequently inhibits the formation and antiviral activity of downstream SGs. To the best of our knowledge, this is the first report showing that NF90 undergoes ubiquitination and also the critical domain (DZF) and critical site (Lys297) of NF90 for Tim-3- and E3 ligase TRIM47-mediated ubiquitination and degradation.

NF90, like the classical PRRs, is considered a novel virus sensor, exerting an important role in host innate immunity against various viral infections, especially negative-sense single-stranded RNA

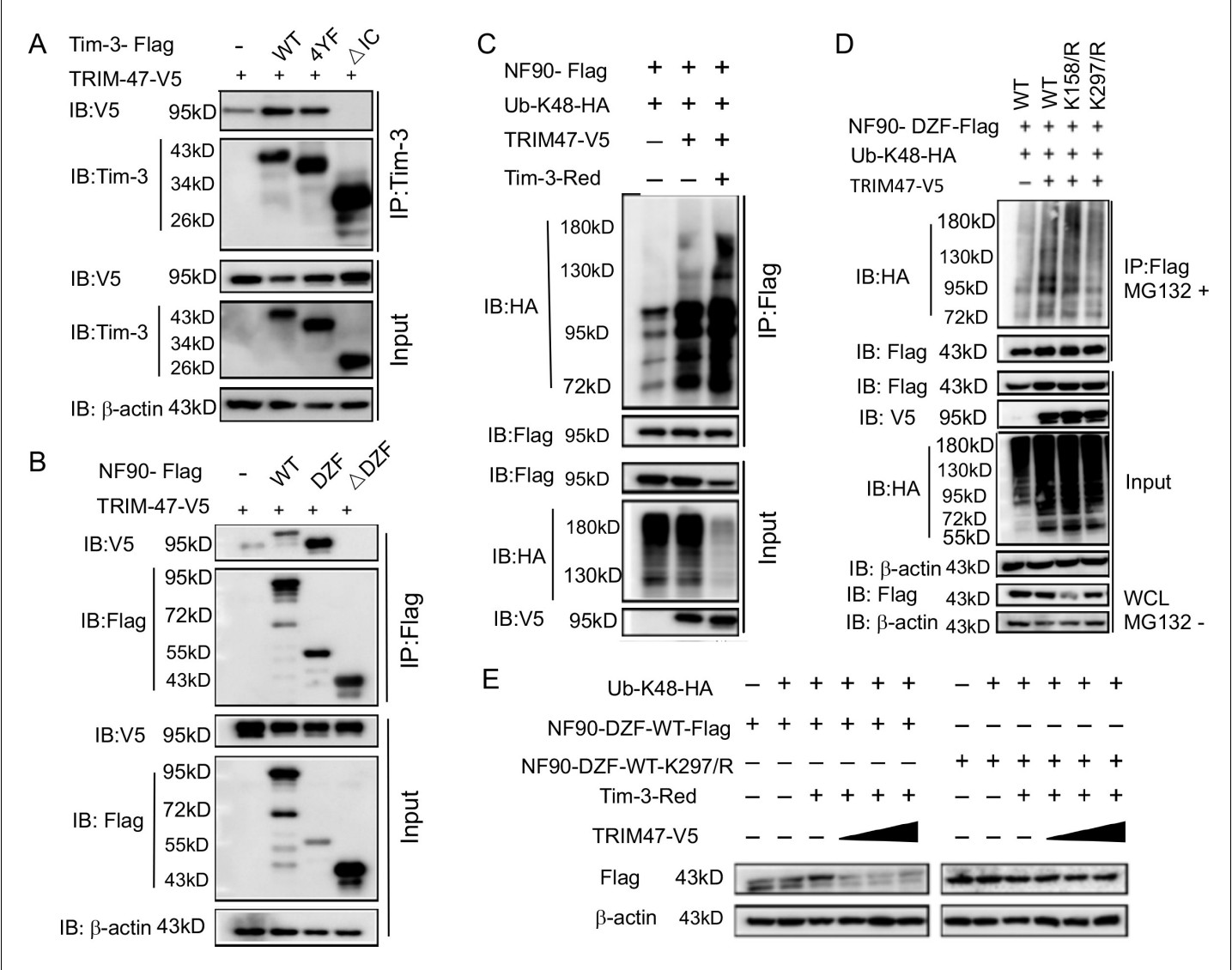

**Figure 5.** Tim-3 recruits TRIM47 to DZF domain of NF90 within which Lys297 is a critical site for TRIM47-mediated K48-linked ubiquitination and degradation of NF90. (A and B) The intracellular domain of Tim-3 and the DZF domain of NF90 interact with TRIM47 respectively. HEK293T cells were transfected with the indicated plasmids for 24 hr and treated with MG132 (20 ug/ml) for 6 hr. The cells were then lysed, immunoprecipitated with Tim-3 or Flag antibody, and detected by western blot using the indicated antibodies. (C) Tim-3 promotes NF90 degradation mediated by E3 ligase TRIM47. HEK293T cells were transfected with plasmids encoding HA-Ub-K48, V5-TRIM47, Flag-NF90, and Red-Tim-3, and treated with MG132 (20 ug/ml) for 6 hr. Cell lysates were then immunoprecipitated with Flag antibody and analyzed by western blot using the indicated antibodies. (D) Residue K297 of NF90 is the major site of TRIM47-mediated K48-linked ubiquitination. Flag-NF90-DZF (WT) or KR mutants, HA-Ub-K48 and V5-TRIM47, were transfected into HEK293T cells for 24 hr. Cells were then treated with MG132 (20 ug/ml) for 6 hr. Cell lysates were analyzed by western blot for K48-linked ubiquitination of NF90. (E) Residue K297 is the decisive site in TRIM47-mediated degradation of NF90. Plasmids encoding Flag-NF90-DZF (WT) or K297R mutants, PDsRed-Tim-3, HA-Ub-K48, and V5-TRIM47, were transfected into HEK293T cells, and cell lysates were examined by western blot for the indicated proteins. Three independent experiments were conducted for all panels.

The online version of this article includes the following source data, source code and figure supplement(s) for figure 5:

**Source code 1.** Lys297 is a critical site for ubiquitination of NF9.
**Source data 1.** Tim-3 interacts with TRIM47 via its intracellular tail.
**Source data 2.** NF90 interacts with TRIM47 via its DZF domain.
**Source data 3.** Tim-3 promotes TRIM47 mediated ubiquitination of NF90.
**Source data 4.** Lys297 is a critical site for TRIM47-mediated ubiquitination of NF90.
**Source data 5.** Lys297 is a critical site for TRIM47-mediated ubiquitination of NF90-DZF.
**Figure supplement 1.** Lys297 of NF90 is a critical site in TRIM47-mediated K48-linked ubiquitination and degradation of NF90.

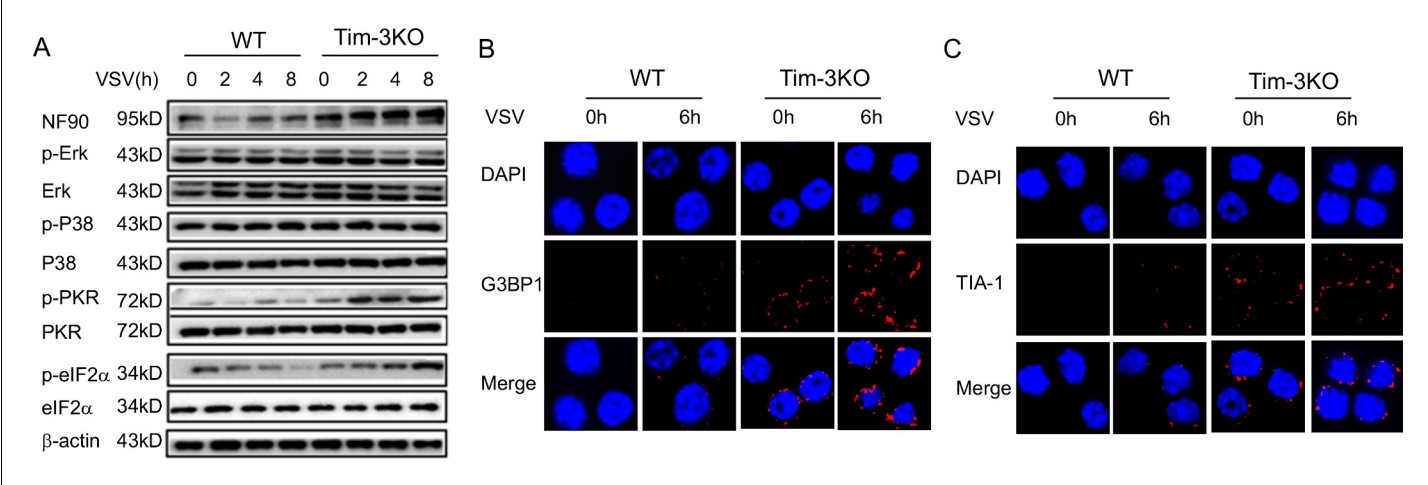

**Figure 6.** Tim-3 selectively inhibits the phosphorylation of PKR and eIF2a and decreases the expression of SG markers G3BP1 and TIA-1 in macrophages. (**A**) Lysates from wild-type (WT) and Tim-3 gene (*Havcr2*) knockout (Tim-3KO) peritoneal macrophages infected with vesicular stomatitis virus (VSV) were analyzed by immunoblotting for the indicated proteins. (**B** and **C**) Peritoneal macrophages isolated for WT or Tim-3KO mice were infected with or without VSV for 6 hr, and then the cells were immunostained with the indicated antibodies and analyzed by fluorescence microscopy. Three independent experiments were conducted for all panels.

The online version of this article includes the following source data, source code and figure supplement(s) for figure 6:

**Source code 1.** TRIM47 knock down increases TIA-1,G3BP1 expression and decrease VSV expression.
**Source data 1.** Tim-3 selectively inhibits the phosphorylation of PKR and eIF2a.
**Source data 2.** Tim-3 increases the expression of SGs markers G3BP1 and TIA-1 in macrophages.
**Figure supplement 1.** TRIM47 knockdown enhances antiviral immunity in macrophages.

viruses (*Li et al., 2016*; *Wang et al., 2009*; *Wen et al., 2014*). One report showed that NF90 is required for an efficient response against VSV infection (*Pfeifer et al., 2008*), but the underlying mechanism has not been clarified. Other reports have shown that NF90 interacted with the VP35 protein of Ebola virus (EBOV) and inhibited EBOV infection through impairing the function of the EBOV transcription/replication complex (*Shabman et al., 2011*). Knockdown of NF90 in indicated cells dramatically promotes EBOV and influenza virus replication, while overexpression of NF90 inhibits or impacts replication of these viruses (*Li et al., 2016*; *Shabman et al., 2011*; *Wang et al., 2009*). When the signaling cascades were investigated, SGs, not the interferon pathway, served as the downstream signaling cascade of the NF90 antiviral pathway (*Shabman et al., 2011*; *Pfeifer et al., 2008*). NF90 promotes (*Meunier et al., 2014*) the assemblage of SG and synergizes with other proteins to exert antiviral immunity (*Wen et al., 2014*). These reports support our findings that NF90 inhibits VSV replication via SGs and further demonstrate the critical role of the NF90-SG signaling pathway during an antiviral immune response.

Tim-3 is an immune checkpoint inhibitor that was initially found to be expressed on activated Th1 cells by binding with its ligand Gal-9 (*Chou et al., 2009*; *Dardalhon et al., 2010*). Tim-3 induces apoptosis and T cell tolerance (*Nakayama et al., 2009*; *Sarraj et al., 2014*). Most investigations focus on the roles of Tim-3 in maintaining T cell exhaustion in immune disorders, tumors, or infectious diseases, which means that this checkpoint inhibitor could be abused. Recently, our works and other published data have focused on the roles of Tim-3 in maintaining the homeostasis of innate immune cells and have demonstrated that the dysregulated Tim-3 on innate immune cells also contributes to the immune evasion of many tumors or pathogens (*Diao et al., 2020*; *Jiang et al., 2016*). Innate immune cells have now attracted much attention for developing new therapeutic strategies against infectious or tumor diseases, in which innate immune cell-expressed immune checkpoint inhibitors are potential targets (*Han et al., 2013*). Here we have found that Tim-3 suppresses the macrophage-mediated antiviral immune response, suggesting that it is a potent therapeutic target for re-boosting innate immunity against viruses. However, as Tim-3 does not possess an obviously classical Immunoreceptor Tyrosine-based Inhibition Motif (ITIM)motif compared with other immune-checkpoint receptors, such as PD-1, CTLA-4 (*Stålhammar et al., 2019*), and Siglec-G (*Chen et al.,*

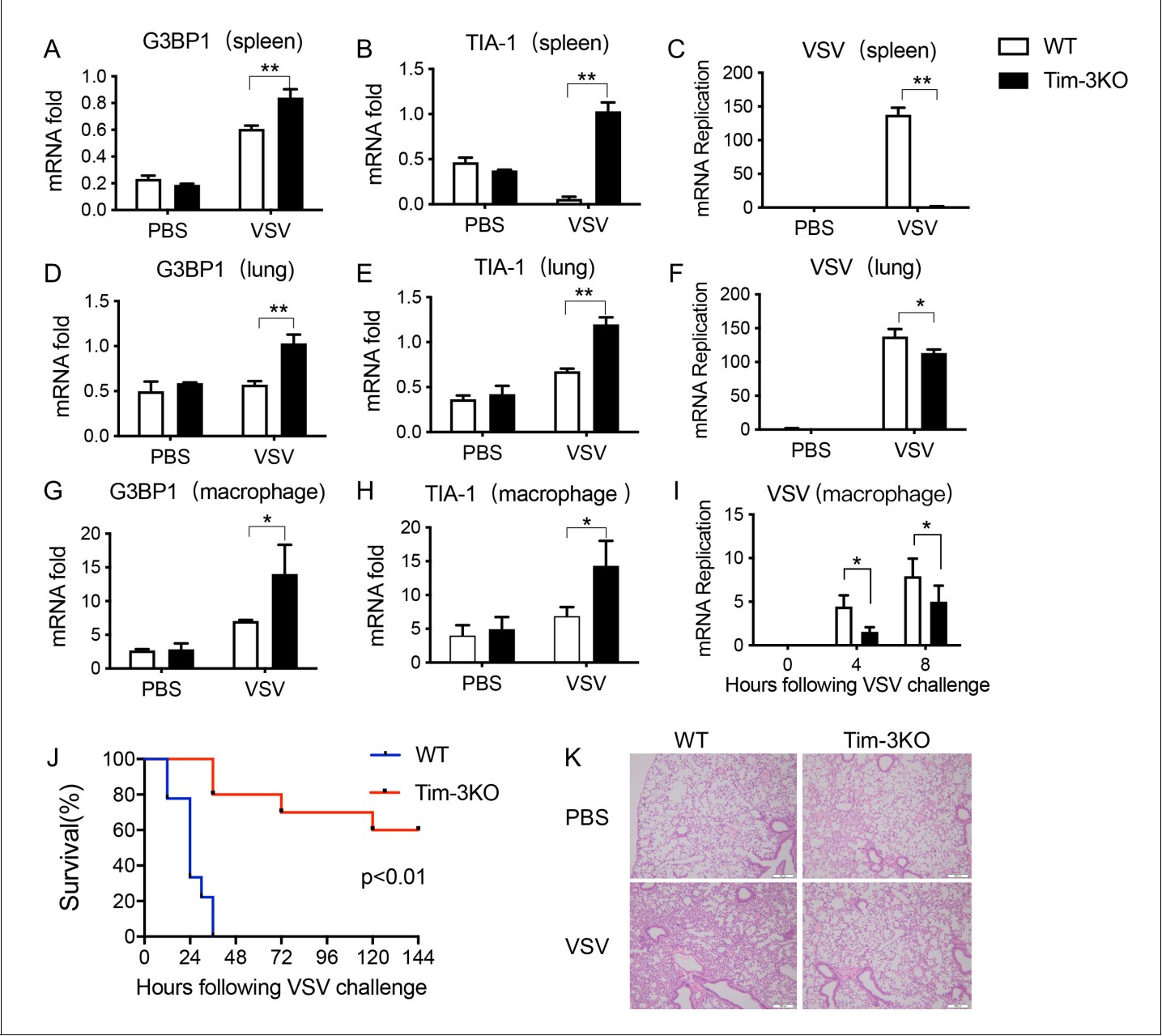

**Figure 7.** Tim-3 deficiency upregulates G3BP1 and TIA-1 and protects mice from VSV infection. (A, B, D, E, G, and H) Detection of mRNA transcription of G3BP1 and TIA-1 in organs and peritoneal macrophages by quantitative reverse transcription PCR (qPCR) after wild-type (WT) and Tim-3KO mice (n = 5 per group) were intraperitoneally injected with vesicular stomatitis virus (VSV) for 24 hr. (C, F, and I) qPCR analysis of VSV loads in organs and peritoneal macrophages after WT and Tim-3KO mice (n = 5 per group) were intraperitoneally injected with VSV for 24 hr. (J) WT and Tim-3KO mice (~7 weeks old) were intraperitoneally injected with VSV (1 × 10$^8$ pfu/g) (n = 10 per group) followed by recording survival of both groups. p<0.01. (K) The lung tissues from WT and Tim-3KO mice (C, F, and I) were stained with hematoxylin and eosin, and their pathology analyzed in response to VSV. The results shown are representative of three independent experiments. *p<0.05; **p<0.01.

The online version of this article includes the following source data for figure 7:

**Source data 1.** Tim-3 knowdown increases G3BP1expression in spleen with VSV infection.
**Source data 2.** Tim-3 knowdown increases TIA-1 expression in spleen with VSV infection.
**Source data 3.** Tim-3 knowdown decreases VSV expression in spleen with infection.
**Source data 4.** Tim-3 knowdown increases G3BP1 expression in lung with VSV infection.
**Source data 5.** Tim-3 knowdown increases TIA-1 expression in lung with VSV infection.
**Source data 6.** Tim-3 knowdown decreases VSV expression in lung with VSV infection.
**Source data 7.** Tim-3 knowdown increases G3BP1 expression in peritoneal macrophages with VSV infection.
**Source data 8.** Tim-3 knowdown increases TIA-1 expression in peritoneal macrophages with VSV infection.

**Source data 9.** Tim-3 knowdown decreases VSV expression in peritoneal macrophages with VSV infection.

**Source data 10.** Tim-3 deficiency protects mice from VSV infection.

**Source data 11.** Tim-3 knowdown attenuated tissue damage.

*2013*), the mechanism by which Tim-3 mediates inhibitory signals remains unclear. *Huang et al., 2015* showed that Tim-3 induces T cell exhaustion via Human leucocyte antigen B associated transcript (BAT) and using CEACAM1 as a coreceptor, and our findings have shown that Tim-3 promotes tumor-prone macrophage polarization by binding to and suppressing the phosphorylation and nuclear translocation of STAT1 (*Jiang et al., 2016*) Further, Tim-3 inhibits TLR4-mediated macrophage activation during sepsis by suppressing NF-kB activation (*Yang et al., 2013*). How Tim-3 works during anti-viral innate immunity remains unclear. The intracellular tail of Tim-3 contains a highly conserved tyrosine-containing src homology 2 (SH2)-binding motif, and tyrosine residues within this motif can be phosphorylated, which is critical for Tim-3 signaling in T cells (*Chen et al., 2013*). In this study, we have identified an increased tyrosine phosphorylation of Tim-3 in macrophages following VSV infection. We have also demonstrated that deletion of Tim-3 tail or mutation of the conserved tyrosines, including Tyr265, Tyr272, Tyr280, and Try281, within Tim-3 tail significantly attenuated the interaction between Tim-3 and NF90. These data demonstrated a new mechanism by which Tim-3 transduces inhibitory signal in anti-viral innate immune responses.

Ubiquitination is one of the most versatile posttranslational modifications for substrates and is indispensable for antiviral infection (*Callis, 2014*). However, whether NF90 undergoes ubiquitination is totally unknown. Structurally, NF90 possesses a DZF in the N-terminal region, which is a symbol of ubiquitination for substrates (*Ji et al., 2018*). Here we have verified our hypothesis and demonstrated that NF90 can be ubiquitinated, a process that is enhanced by Tim-3 in macrophages following VSV infection. As speculated, Tim-3 mainly promotes the ubiquitination of the DZF domain of NF90. To the best of our knowledge, this is the first report demonstrating that NF90 undergoes ubiquitination. Further, we have also found that Lys297, a highly conserved residue among different isoforms of NF90, plays a critical role in the K-48-linked ubiquitination of NF90.

When the candidates accounting for NF90 ubiquitination were examined, we focused on TRIM47 as it got the highest scores and the highest number of matched peptides among NF90-interacting proteins, and structurally, TRIM47 contains a RING finger domain in the N-terminus, which may contribute to ubiquitin modification (*Ji et al., 2018*). In addition, TRIM family proteins play important roles in many biological processes, such as cell cycle regulation and viral response (*Wang et al., 2015*). As discussed above, the conserved tyrosines within the Tim-3 tail could form an SH2 (SRC homology 2)-binding domain (*Rangachari et al., 2012*; *van de Weyer et al., 2006*; *Jiang et al., 2016*; *Lee et al., 2011*). We posit that Tim-3 may recruit TRIM47 using its SH2-binding domain as the tyrosines are phosphorylated following VSV infection. A recent report showing that Siglec-G triggers downstream signaling by recruiting SHP-2 (*Chen et al., 2013*) supports this hypothesis. Here we have demonstrated that TRIM47 dose-dependently promotes the proteasome-dependent degradation of NF90 (*Figure 4B*). However, to the best of our knowledge, no well-defined ligase-dead TRIM47 construct(s) has been reported so far. The mechanisms by which TRIM47 promotes NF90 degradation remain to be determined in the future. Interestingly, our data showed that Tim-3 enhances the expression and ubiquitination of TRIM47. This is also the first report showing the ubiquitination modification of TRIM47. However, the mechanisms of Tim-3-enhanced TRIM47 ubiquitination and whether the enhanced ubiquitination of TRIM47 by Tim-3 accounts for the increased TRIM47 expression remain to be determined.

In summary, we have verified that Tim-3 is specifically upregulated following VSV infection and inhibits NF90 signaling pathway in macrophages. Intracellularly, Tim-3 promotes TRIM47-mediated ubiquitination and degradation of NF90. In VSV-infected models, Tim-3 signaling inhibits the formation and the activity of the NF90 downstream SGs. To the best of our knowledge, this is the first report demonstrating the ubiquitination modification of NF90. These findings provide a novel mechanism of Tim-3-mediated infection tolerance, which has implications in antiviral applications.

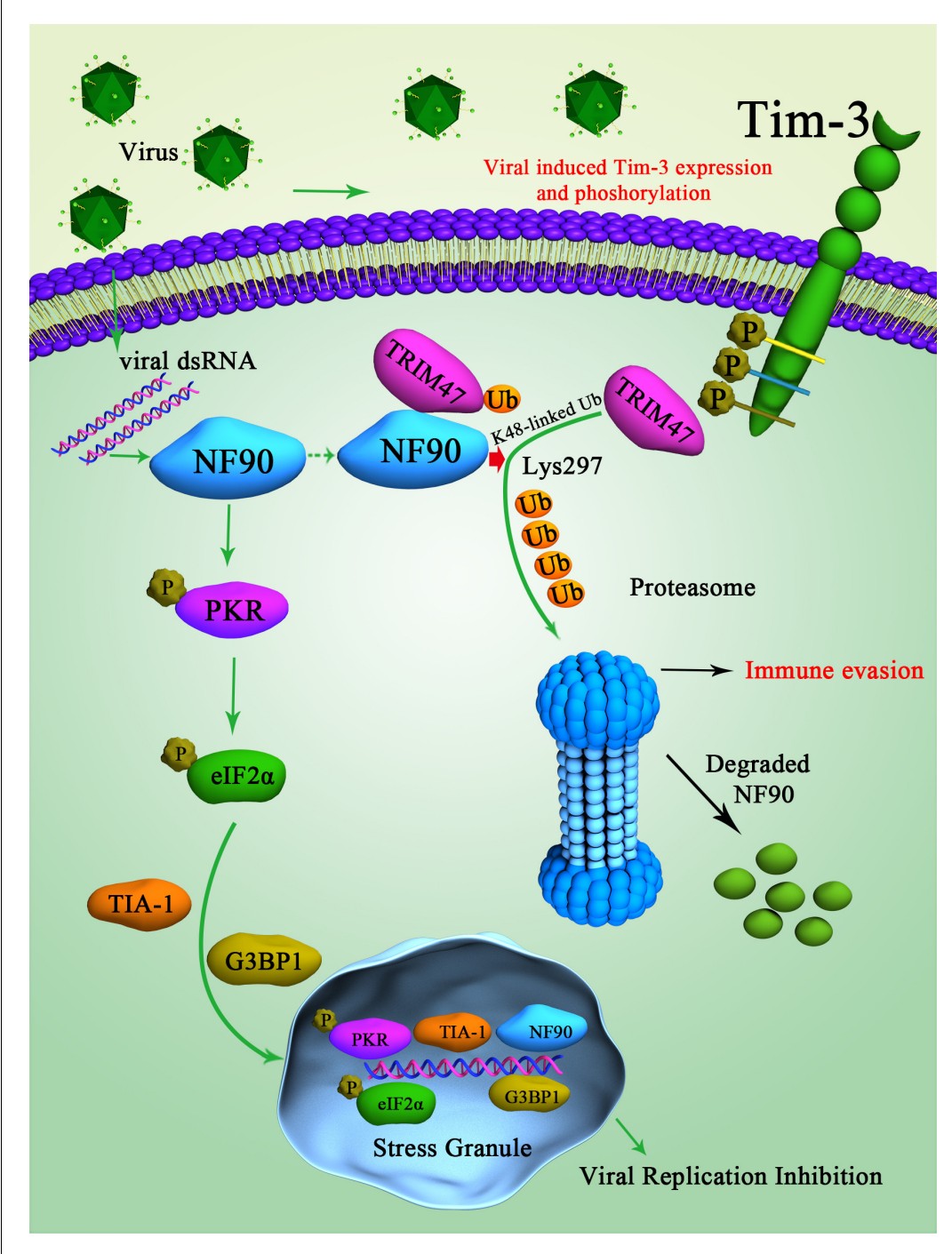

**Figure 8.** Schematic diagram of how Tim-3 inhibits NF90-SG pathway in macrophages during infection. Upon vesicular stomatitis virus (VSV) infection, Tim-3 is activated and upregulated. The activation of Tim-3 in turn recruited the E3 ubiquitin ligase TRIM47 to the zinc finger domain of NF90 and initiated a proteasome-dependent degradation of NF90 via K48-linked ubiquitination at Lys297. The negative regulation of NF90 by Tim-3 blocked the RNA virus and triggered NF90-SG-mediated antiviral immunity and finally led to virus immune evasion.

## Materials and methods

### Mice

The Tim-3-TG mice were produced and fed as described previously (*Jiang et al., 2016*). The *Havcr2*^f/f^ (Tim-3KO) mice (C57BL/6) used in this study were generated in the Transgenic Core Facility of Cyagen Biosciences Inc, Guangzhou, China. EIIa-cre mice carry a cre transgene under the control of the adenovirus EIIa promoter that targets expression of Cre recombinase to the early mouse embryo and are useful for the germline deletion of Loxp-flanked genes. EIIa-cre knock-in mice (C57BL/6) were a gift from Dr. Haitao Wu (Institute of Beijing Institute of Basic Medical Sciences, Beijing, China). Tim-3KO (*Havcr2*^f/f^ *EIIa*^cre/+^) mice (C57BL/6) were generated by mating Tim-3-flox with EIIa-cre mice, and the genotype of Tim-3KO mice (*Havcr2*) was detected with a primer set as *mHavcr2*-Forward: 5'-CCAATTGGGTTCTACTATAAAGCCTTG-3', *mHavcr2*-Reverse1: 5'-AAGTTGA-GAGTTCTGGGATTACAGG-3', and *mHavcr2*-Reverse2: 5'-ATACTTGCTTCAGTGGCTCGCGA-3' (*Figure 1—figure supplement 1*). Wild-type C57BL/6 mice and the aforementioned Tim-3-TG and Tim-3KO mice were bred in specific pathogen-free conditions. The protocol was approved by the Ethics Committee of Animal Experiments of the Beijing Institute of Brain Science (IACUC-DWZX-2018–645). All efforts were made to minimize suffering. Major procedures were blinded.

### Cells and reagents

The RAW264.7 and HEK293T cell lines (ATCC) were maintained in Dulbecco's modified Eagle's medium (Gibco) supplemented with 10% fetal bovine serum (Gibco). The identity of cell lines has been authenticated using short tandem repeat(STR) profiling and no mycoplasma contamination was found. Peritoneal macrophages were prepared as described (*Wang et al., 2016*). For stable transfection of NC shRNA or Tim-3 shRNA, RAW264.7 macrophages were transfected with Lipofectamine 2000 reagents (Invitrogen; 11668019) and then selected with 1000 ng/ml G418 (Invitrogen; 10131027), which were purchased from Invitrogen. VSV was obtained and cultivated as described (*Chen et al., 2013*; *Jiang et al., 2016*). MG132 was purchased from Selleckchem (S2619) and used at a final concentration of 20 µM.

### Plasmids and antibodies

The flag-tagged full-length NF90, NF90 mutation plasmids (NF90-DZF and NF90-ΔDZF), V5-tagged full-length NF90, as well as HA-tagged ubiquitin, were constructed into pcDNA3.1 (+)-Flag and pcDNA3.1 (+)-V5 eukaryotic expression vectors, respectively. Recombinant vectors encoding WT or mutant human-specific Tim-3 were constructed by PCR-based amplification of complementary DNA (cDNA) from human U937 cells and then subcloned into the pcDNA3.1 (+) eukaryotic expression vector, with Flag, Myc, and HA tags, respectively. Full-length NF90-GFP and full-length Tim-3-RFP fluorescence plasmids were cloned into PEGFP1-N1 and PDsRed1-N1 eukaryotic expression vectors, respectively. TRIM47 full-length for V5-tag plasmids were constructed into the pcDNA3.1 (+)-V5 eukaryotic expression vector. Antibodies to Tim-3 (D3M9R, mouse-specific), Tim-3 (D5D5R, human-specific), eIF2α (D7D3), p-eIF2α (D9G8), Ub-K48 (D9D5), P38 (8690), p-P38 (9215), ERK (4348), and p-ERK (8544) were obtained from Cell Signaling Technology. An antibody to p-PKR (GTX32348) was purchased from GeneTex. Antibodies to β-actin, anti-Flag-Tag (CW0287), anti-HA-Tag (CW0092), anti-V5-Tag (CW0094), and anti-RFP (CW0253) were from purchased from CWBIO (China). Antibodies to anti-V5-Tag (ab9116) and anti-HA-Tag (ab9110) for immunoprecipitation were obtained from Abcam. Antibodies to anti-Flag-Tag (F1804) for immunoprecipitation were obtained from Sigma. Antibodies to Ub (ab7780), ILF3/(NF90) (ab92355), PKR (ab184257), G3BP (ab181150), and TIA-1 (ab140595) were obtained from Abcam. The antibody to TRIM47 (BC017299) was obtained from Thermo (PA5-50892), and the antibody to Tim-3 (A2516) for western blot was obtained from Abclonal.

### Western blot analysis

Western blots were performed as described previously (*Yang et al., 2013*). Briefly, cells were lysed with lysis buffer (1% Triton X-100, 20 mM Tris-HCl, pH 8.0, 250 mM NaCl, 3 mM ethylenediaminetetraacetic acid (EDTA), pH 8.0), 3 mM ethylene glycol tetraacetic acid (pH 8.0) with the pH adjusted to 7.6, and complete protease inhibitor cocktail (Roche; pH 7.5) on ice for 30 min. Lysates were

eluted by boiling for 10 min with 5× sample buffer (100 mM Tris-HCl, pH 6.8, 2% sodium dodecyl sulfate (SDS), 10% glycerol, 0.1% bromophenol blue, 1% β-mercaptoethanol) and were separated by 10% sodium dodecyl sulphate–polyacrylamide gel electrophoresis (SDS/PAGE), followed by examination of expression levels of the indicated proteins: phospho-eIF2α, eIF2α, phospho-PKR, PKR, total protein of NF90, and the levels of phospho-ERK and phospho-p38. β-actin served as an internal control.

## Viral load

RAW264.7 cell supernatants were collected 3 or 6 hr after VSV infection. Serial tenfold dilutions of the supernatant were added to HEK293T cell monolayers in 96-well plates and they were incubated for 72 hr at 37°C. Endpoints of cytopathic effect were observed, and 50% tissue cell infectious dose (TCID50) was determined using the Reed-Muench method.

## Co-immunoprecipitation

Cells were collected 24 hr after transfection and lysed in lysis buffer (1% NP-40, 20 mM Tris-HCl, 150 mM NaCl, 5 mM EDTA, 1 mM $Na_3VO_4$, 0.25% sodium deoxycholic acid, and complete protease inhibitor cocktail (Roche), pH 7.5) on ice for 30 min. After centrifugation for 15 min at 12,000 r (11800 × g), 4°C, the supernatants were collected and incubated with Protein A/G Sepharose beads (SC-2003; Santa Cruz) coupled to specific antibodies overnight at 4°C. The next day, beads were washed three times with high-salt wash buffer (1% Triton X-100, 20 mM Tris-HCl, 500 mM NaCl, 10% glycerol, 2 mM EDTA, 1 mM $Na_3VO_4$, and complete protease inhibitor cocktail (Roche), pH 7.5) and another three times with low-salt wash buffer (1% Triton X-100, 20 mM Tris-HCl, 150 mM NaCl, 10% glycerol, 2 mM EDTA, 1 mM $Na_3VO_4$, and complete protease inhibitor cocktail (Roche), pH 7.5). Lysates were eluted by boiling for 10 min with 5× sample buffer (as indicated). Precipitates were fractionated by SDS/PAGE with appropriate concentrations and western blot was performed as described above.

## Ubiquitination assays

For analysis of the ubiquitination of NF90 in HEK293T cells, plasmids encoding Flag-NF90 and HA-Ub-K48 were transfected into HEK293T cells for 24 hr and were treated with MG132 (20 μM) for 6 hr before harvesting. Cells were lysed with immunoprecipitation lysis buffer (1% NP-40, 20 mM Tris-HCl, 150 mM NaCl, 5 mM EDTA, 1 mM $Na_3VO_4$, 0.5% sodium deoxycholic acid, and complete protease inhibitor cocktail (Roche), pH 7.5), and then the whole-cell lysates were immunoprecipitated with an antibody to Flag tag (F1804), followed by analysis of ubiquitination of NF90 with an antibody to HA tag. Precipitates were fractionated by SDS/PAGE with appropriate concentrations (as indicated).

## Pathology and survival assays

For survival assays, ~6-week-old WT and Tim-3KO mice were given intraperitoneal injection with VSV ($1 \times 10^8$ pfu/g) (n = 9 per group). To detect the pathology of WT and Tim-3KO mice in response to VSV, the hematoxylin and eosin staining of lung sections was examined 24 hr after infecting.

## Mass spectrometry

Plasmids encoding full-length Tim-3 were transfected into HEK293T cells for 24 hr and cell lysates were immunoprecipitated with an antibody to Tim-3 (D5D5R #45208). Mass spectrometry was used to identify Tim-3-interacting proteins.

## qPCR and RNAi knockdown

Gene expression was analyzed by three-step quantitative reverse transcription PCR (qPCR). Total RNA was extracted from mouse macrophages using TRI reagent (Sigma). Following the manufacturer's instructions, RNA was reverse-transcribed in a 20 μl reaction volume (42°C, 30 min; 95°C, 5 min) using a QuantiTect Reverse Transcription Kit (Qiagen, Valencia, CA). cDNA was then amplified using a SYBR Green I Master mix (Roche, Basel, Switzerland) and a Light Cycler 480 PCR system (Roche). All tests were carried out in duplicate reaction mixtures in 96-well plates. The relative expression of the gene of interest was determined using the $2^{-\Delta\Delta Ct}$ method, with 18S ribosomal

mRNA (18S) as the internal control. The primers used for qPCR are listed in *Figure 1—figure supplement 1*.

## Immunofluorescence microscopy

Peritoneal macrophage cells grown on specific confocal dishes were infected with VSV for 6 hr. Thereafter, the cells were fixed in 4% paraformaldehyde, permeabilized with 1× Tris-buffered saline with Tween 20 (TBST) with 0.5% Triton X-100 for 10 min, then incubated in blocking buffer (1× TBST with 3% bovine serum albumin), and then incubated with primary antibodies overnight at 4°C. The cells were washed three times with 1× TBST and incubated with secondary antibodies conjugated to Alexa Fluor 594 for 2 hr at room temperature (ZSGB-BIO,China) and washed again three times with 1× TBST. Nuclei were stained with 4′,6-diamidino-2-phenylindole (SouthernBiotech; 0100–20) for 10 min, and then the dishes were visualized with a C1-si confocal fluorescence microscope (Nikon Instruments, Tokyo, Japan). The overlap coefficient, area/diameter of SG dots, and line intensity were calculated with Image-Pro Plus Software (Media Cybernetics, Rockville, MD,).

## Statistical analysis

The significance of difference between groups was determined by two-tailed Student's t-test and two-way analysis of variance test. For the mouse survival study, Kaplan–Meier survival curves were generated and analyzed for statistical significance with GraphPad Prism 6.0. p-values <0.05 were considered statistically significant.

## Acknowledgements

We thank Prof. Minghong Jiang, Institute of Basic Medical Sciences, Peking Union Medical College, Chinese Academy of Medical Sciences, Beijing, China, for critical reading. This work was supported by the National Natural Sciences Foundation of China (grant nos. 81771684 and 81971473) and the Beijing Natural Sciences Foundation (grant no. 7192145).

## Additional information

### Funding

| Funder | Grant reference number | Author |
| --- | --- | --- |
| National Natural Science Foundation of China | 81771684 | Gencheng Han |
| National Natural Science Foundation of China | 81971473 | Gencheng Han |
| Beijing Natural Science Foundation | 7192145 | Gencheng Han |

The funders had no role in study design, data collection and interpretation, or the decision to submit the work for publication.

### Author contributions

Shuaijie Dou, Formal analysis, Investigation, Methodology; Guoxian Li, Yang Gao, Investigation, Methodology; Ge Li, Data curation, Formal analysis, Project administration; Chunmei Hou, Methodology; Yang Zheng, Formal analysis, Methodology; Lili Tang, Investigation, Project administration; Rongliang Mo, Yuxiang Li, Data curation, Investigation; Renxi Wang, Conceptualization, Project administration; Beifen Shen, Conceptualization, Funding acquisition; Jun Zhang, Data curation, Investigation, Project administration; Gencheng Han, Formal analysis, Funding acquisition, Validation, Project administration, Writing - review and editing

### Author ORCIDs

Gencheng Han (iD) https://orcid.org/0000-0003-1408-878X

### Ethics

Animal experimentation: The protocol was approved by the Ethics Committee of Animal Experiments of the Beijing Institute of Brain Sciences (IACUC-DWZX-2018-645). All efforts were made to minimize suffering.

### Decision letter and Author response

Decision letter https://doi.org/10.7554/eLife.66501.sa1

Author response https://doi.org/10.7554/eLife.66501.sa2

## Additional files

### Supplementary files

- Supplementary file 1. Sequences of the primers used for PCR.

- Transparent reporting form

### Data availability

All data generated or analysed during this study are included in the manuscript and supporting files.

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
