## [Decision Letter]

**Acceptance summary:**

Stress granules are a critical component in the defense against certain viruses. The mechanisms regulating stress granules induced but the NF90 pathway are not well defined. Here, the authors use biochemical, cell-based, and in vivo approaches to make the novel discovery that the immune checkpoint inhibitor Tim3 functions as a negative regulator of NF90-mediated antiviral responses.

**Decision letter after peer review:**

Thank you for submitting your article "Ubiquitination and degradation of NF90 by Tim-3 inhibits antiviral innate immunity" for consideration by *eLife*. Your article has been reviewed by 2 peer reviewers, one of whom is a member of our Board of Reviewing Editors, and the evaluation has been overseen by Satyajit Rath as the Senior Editor. The reviewers have opted to remain anonymous.

Essential revisions:

Reviewers request additional experimentation to substantiate Tim3-mediated effects on SG formation, along the lines of comment #4 from Reviewer #1 and comment #6 from Reviewer #2. The remaining comments can be addressed either experimentally or through additional discussion of caveats and limitations in Discussion. Please provide a detailed response of how all points from both reviewers has been considered.

*Reviewer #1:*

In this study, the authors sought to uncover mechanism regulating NF90-induced antiviral pathways. They initially discovered that the checkpoint protein Tim3 was both induced upon viral infection and that loss of Tim3 is protective against the infection. Mechanistically, the authors used biochemical and cell-based approaches to discover that Tim3 binds NF90 and promotes its degradation via ubiquitination mediated by the E3 ligase TRIM47. The authors further assess the functional consequence of Tim3-mediated inhibition of NF90. As NF90 has known roles in stress granule formation, the authors demonstrated that loss of Tim3 expression is associated with increased NF90, which correlated with increased phosphorylation of PKR and eIF2a and induction of stress granule markers. To demonstrate biological relevance, the authors reveal that mice lacking Tim3 are more resistant to VSV infection.

Strengths include diverse experimental approaches, including in vivo work, to identify and characterize the TIM3-TRIM47-NF90 axis. Weaknesses include lack of a few key experiments that will help support what is already a strong study. The impact of the work is fundamental new insight into the pathways regulating an important mechanism of viral control.

1. What are the effects of Tim3 on VSV titers? RNA is not always a reliable surrogate for infectious virus production.

2. In Figure 3D, why does the full-length protein not appear to be ubiquitinated while the DZF only construct is strongly ubiquitinated?

3. For the TRIM47 experiments, can the authors generate a ligase-dead TRIM47 to firmly conclude that it is the E3 ligase activity of this protein, and not some other cellular interactor that mediates NF90 degradation. Additionally, can the authors discuss why K63-Ub was examined in the context of TRIM47 as opposed to K48, etc? The details of this specific choice were not clearly outlined in the text.

4. In both the cell culture models and the in vivo studies, direct visualization of stress granules by immunofluorescence in Tim3+ and Tim- cells/mice would be more compelling that SG RNA marker induction, as it's not clear what threshold of mRNA is needed to actually form the SGs. Since the authors are making the claim that the viral phenotypes are mediated via by Tim3-dependent effects on NF90-induced SG formation, some demonstration of SG formation in the cell/tissue would provide stronger evidence for the mechanism.

*Reviewer #2:*

In the present manuscript, the authors provide data on a cross-talk between the immune checkpoint molecule Tim-3 in macrophages and the viral sensor NF90 in the context of VSV infection. In particular, they demonstrate (by using cell lines and primary cells) that VSV infection leads to increased expression and activation of Tim-3, Tim-3 interacts with NF90 via its cytoplasmic tail and enhances NF90 ubiquitination by recruitment of TRIM47.Subsequently, NF90 is degraded and VSV replication is increased. Thus, Tim-3 can mediate viral escape in VSV infection. With this, the authors provide a novel viral escape mechanism involving the previously described immune checkpoint molecule Tim-3 and NF90 in macrophages. Although the authors thoroughly identified key players within this viral escape pathway future studies are required to comprehensively resolve subsequent steps within this pathway. The manuscript is well written and structured and the data appear reliable.

Specific suggestions to the authors:

1. Figure 1: Quantification of the Tim-3 protein expression (either by densitometric WB analysis or flow cytometry) as well as a prolonged time-course analyzing the expression (mRNA and protein) in the same experimental system (RAW264.7 versus primary macrophages) would clarify the kinetics and strength of Tim-3 regulation by VSV. This will add insights into the potential of NF90 regulation by Tim-3 during VSV infection.

2. Line 138: The described data of NF90 ubiquitination in primary mouse macrophages should be provided and not just stated "data not shown".

3. Please indicate the exact number of independent experiments that were performed for each data set.

4. Line 602: Typo F instead of E.

5. Figure 6: The data showing the effects on SGs would be substantiated by confirmation of the differential expression of G3BP-1 and TIA-1 on protein level.

---

## [Author Response]

Essential revisions:Reviewers request additional experimentation to substantiate Tim3-mediated effects on SG formation, along the lines of comment #4 from Reviewer #1 and comment #6 from Reviewer #2. The remaining comments can be addressed either experimentally or through additional discussion of caveats and limitations in Discussion. Please provide a detailed response of how all points from both reviewers has been considered.

Thanks for these comments. We further substantiated the effects of Tim-3 on SG formation by immunofluorescence (seen below and also in revised Figure 6B and C). And according to the advice of the reviewers, we also provided some other new data and additional discussions.

Reviewer #1:[…] Weaknesses include lack of a few key experiments that will help support what is already a strong study. The impact of the work is fundamental new insight into the pathways regulating an important mechanism of viral control.1. What are the effects of Tim3 on VSV titers? RNA is not always a reliable surrogate for infectious virus production.

We repeated the experiments and found that when macrophages were infected with VSV, Tim-3 silence or knockdown lead to decreased VSV titers (seen below and also in revised Figure 1D and E).

2. In Figure 3D, why does the full-length protein not appear to be ubiquitinated while the DZF only construct is strongly ubiquitinated?

Thanks for this comment. The image in the third line of Figure 3B and in the third line of 3D show a slight increase of NF90 ubiquitination when K48-Ub is co-transfected. However, co-transfection of Tim-3 with NF90 significantly increased the ubiquitination of NF90 (the last line in Figure 3B).

Here we demonstrate that NF90 can be ubiquitinated by Tim-3 and identified that the DZF domain is the major ubiquitination site. “The reason why DZF only construct is strongly ubiquitinated may lie in conformational hindrance, which will be investigated in the future.

In the revised manuscript, we added the following description:

“The reason why the DZF only construct is strongly ubiquitinated compare to the full-length construct of NF90 may lie in the conformational hindrance in NF90 at a steady state, which need to be demonstrated in the future.”

3. For the TRIM47 experiments, can the authors generate a ligase-dead TRIM47 to firmly conclude that it is the E3 ligase activity of this protein, and not some other cellular interactor that mediates NF90 degradation. Additionally, can the authors discuss why K63-Ub was examined in the context of TRIM47 as opposed to K48, etc? The details of this specific choice were not clearly outlined in the text.

Thanks for these comments. The E3 ligase activity of TRIM47 was identified recently. To the best of our knowledge, no well-defined ligase-dead TRIM47 construct(s) have been reported yet. Here we demonstrated that TRIM47 dose-dependently promotes the proteasomal dependent degradation of NF90 (Figure 4B).

According to the advice of the reviewers, we addressed the limitations in the discussion part as follows: “Here we demonstrated that TRIM47 dose-dependently promotes the proteasomal dependent degradation of NF90 (Figure 4B). However, to the best of our knowledge, no well-defined ligase-dead TRIM47 construct(s) have been reported so far. The detail mechanisms by which TRIM47 promotes NF90 degradation remain to be determined in the future.

The data in Figure 4D and E shows that Tim-3 signaling enhances TRIM47 expression. To find the mechanisms by which Tim-3 promotes TRIM47, we hypothesized and examined K63-Ub rather than K48-Ub mediated ubiquitination, as the ubiquitination by K48-Ub leads to protein degradation.

4. In both the cell culture models and the in vivo studies, direct visualization of stress granules by immunofluorescence in Tim3+ and Tim- cells/mice would be more compelling that SG RNA marker induction, as it's not clear what threshold of mRNA is needed to actually form the SGs. Since the authors are making the claim that the viral phenotypes are mediated via by Tim3-dependent effects on NF90-induced SG formation, some demonstration of SG formation in the cell/tissue would provide stronger evidence for the mechanism.

Thanks for these comments. We provided new data showing the effects of Tim-3 on SG formation by immunofluorescence. (Revised Figure 6B and C).

Reviewer #2:[…] Specific suggestions to the authors:1. Figure 1: Quantification of the Tim-3 protein expression (either by densitometric WB analysis or flow cytometry) as well as a prolonged time-course analyzing the expression (mRNA and protein) in the same experimental system (RAW264.7 versus primary macrophages) would clarify the kinetics and strength of Tim-3 regulation by VSV. This will add insights into the potential of NF90 regulation by Tim-3 during VSV infection.

Thanks. We quantified the effects of VSV on Tim-3 mRNA and protein expression using the same experimental system (primary macrophages) according to the advice of this reviewer. We have analyzed the effects of VSV on Tim-3 protein expression in peritoneal macrophages on a prolonged time-course before and they are now shown in Revised Figure 1.

2. Line 138: The described data of NF90 ubiquitination in primary mouse macrophages should be provided and not just stated "data not shown".

Thanks for these comments. In revised Figure 3, we provided the image of K48- linked ubiquitination of NF90. And the “input” image was added to supplemental Figure S2.

3. Please indicate the exact number of independent experiments that were performed for each data set.

OK, the number of independent experiments was added in each figure.

4. Line 602: Typo F instead of E.

Thanks, it is now corrected.

5. Figure 6: The data showing the effects on SGs would be substantiated by confirmation of the differential expression of G3BP-1 and TIA-1 on protein level.

Thanks. The differential expression of G3BP-1 and TIA-1 on protein level are demonstrated by immunofluorescence, and they are now shown in revised Figure 6B and C.